# Two FGFRL-Wnt circuits organize the planarian anteroposterior axis

**M Lucila Scimone**[1,2,3†], **Lauren E Cote**[1,2,3†], **Travis Rogers**[1,2,3], **Peter W Reddien**[1,2,3*]

[1]Whitehead Institute for Biomedical Research, Cambridge, United States; [2]Department of Biology, Massachusetts Institute of Technology, Cambridge, United States; [3]Howard Hughes Medical Institute, Massachusetts Institute of Technology, Cambridge, United States

**Abstract** How positional information instructs adult tissue maintenance is poorly understood. Planarians undergo whole-body regeneration and tissue turnover, providing a model for adult positional information studies. Genes encoding secreted and transmembrane components of multiple developmental pathways are predominantly expressed in planarian muscle cells. Several of these genes regulate regional identity, consistent with muscle harboring positional information. Here, single-cell RNA-sequencing of 115 muscle cells from distinct anterior-posterior regions identified 44 regionally expressed genes, including multiple *Wnt* and *ndk/FGF receptor-like (ndl/FGFRL)* genes. Two distinct FGFRL-Wnt circuits, involving juxtaposed anterior *FGFRL* and posterior *Wnt* expression domains, controlled planarian head and trunk patterning. *ndl-3* and *wntP-2* inhibition expanded the trunk, forming ectopic mouths and secondary pharynges, which independently extended and ingested food. *fz5/8-4* inhibition, like that of *ndk* and *wntA*, caused posterior brain expansion and ectopic eye formation. Our results suggest that FGFRL-Wnt circuits operate within a body-wide coordinate system to control adult axial positioning.

**\*For correspondence:** reddien@wi.mit.edu

†These authors contributed equally to this work

**Competing interests:** The authors declare that no competing interests exist.

## Introduction

Adult animals replace cells during tissue turnover and, in many cases, regeneration. How animals specify and maintain regional tissue identity during these processes is poorly understood. Planarians can regenerate any missing body part and replace aged tissues during homeostasis, presenting a powerful system for identifying adult positional information mechanisms (*Reddien and Sánchez Alvarado, 2004*; *Reddien, 2011*).

Planarian regeneration requires an abundant population of dividing cells called neoblasts that includes pluripotent stem cells (*Wagner et al., 2011*; *Rink, 2013*; *Reddien, 2013*). Accordingly, many genes required for regeneration are required for neoblast biology. However, some phenotypes associated with gene inhibition do not impact the capacity of animals to regenerate, but instead affect the outcome of regeneration, suggestive of a role for such genes in providing positional information. For example, inhibition of components of the Wnt signaling pathway causes regeneration of heads in place of tails, generating two-headed animals with heads facing opposing directions (*Petersen and Reddien, 2008*; *Gurley et al., 2008*; *Iglesias et al., 2008*). Neoblasts are also constantly utilized for the replacement of differentiated cells during natural tissue turnover. Several striking planarian phenotypes associated with altered regional tissue identity during tissue turnover have also been identified, including hypercephalized (Wnt-signaling inhibition) (*Petersen and Reddien, 2008*; *Gurley et al., 2008*; *Iglesias et al., 2008*) and ventralized (BMP-signaling inhibition) (*Reddien et al., 2007*; *Molina et al., 2007*; *Orii and Watanabe, 2007*) planarians. Reminiscent of the roles of Wnt and Bmp in planarian regeneration and tissue turnover, Wnt regulates anterior-posterior (AP) axis development (*Petersen and Reddien, 2009b*; *Niehrs, 2010*) and

**eLife digest** Some animals can regrow tissues that have been amputated. A group of flatworms called planarians are often used as a model to study the regeneration process because they are able to restore any lost tissue or even an entire animal from tiny pieces of the body. For regeneration to be successful, it is important to ensure that the new tissues form in the correct locations in the body.

The planarian body is divided into three main parts: head, trunk and tail. Several gene products involved in specifying what tissues regenerate are made by muscle cells along the planarian body. Some of the genes are involved in mechanisms that allow cells to communicate with each other, such as the Wnt signaling pathway. These genes could form a coordinated system to control regeneration, but their precise roles remain poorly understood.

Two groups of researchers have now independently identified genes that provide cells with information about their location in the flatworm body. Scimone, Cote et al. used a technique called RNA sequencing in individual muscle cells to identify 44 genes that have different levels of expression across the head, trunk and tail regions. These genes included multiple components of the Wnt signaling pathway and others that encode members of the FGFRL family of signaling proteins.

Further experiments revealed two distinct sets of genes, or "gene circuits", that provide information to correctly position tissues in the head and trunk regions of the worm. For example, inhibiting the activity of the *wntP-2* or *ndl-3* genes increased the size of the trunk of the worms and caused extra mouths and pharynges (muscular organ used for eating) to form. On the other hand, blocking the activity of genes in the other gene circuit caused the brain to expand and extra eyes to form.

Another study by Lander and Petersen found that *wntP-2 and ndl-3* act with another gene called *ptk7*, which encodes another component of the Wnt signaling pathway. Together these findings suggest that the Wnt-FGFRL circuits act in a body-wide system that co-ordinates where and which new tissues form during regeneration. A future challenge is to find out how the genes identified in these studies interact and how the cells of the animal interpret this information to properly regenerate missing tissues.

---

Bmp regulates dorsal-ventral (DV) axis development (*De Robertis and Sasai, 1996*) in many animal phyla.

Many receptors, ligands, and secreted inhibitors belonging to key pathways that regulate development in many organisms, such as BMP and Wnt pathways are constitutively expressed in a regionalized manner across adult planarian body axes (*Reddien, 2011*). Interestingly, these genes are predominantly expressed together in the same planarian tissue, the body-wall muscle (*Witchley et al., 2013*). Expression patterns of these genes can change dynamically following injury (*Petersen and Reddien, 2008*; *Petersen and Reddien, 2009a*; *Gurley et al., 2010*; *Witchley et al., 2013*), and some of these changes can occur in existing muscle cells in the absence of neoblasts (*Witchley et al., 2013*). Body-wall muscle is distributed peripherally around the entire planarian body, and the known expression domains of candidate patterning molecules in muscle broadly span the AP, DV, and medial-lateral (ML) body axes, raising the possibility that muscle provides a body-wide coordinate system of positional information that controls regional tissue identity in tissue turnover and regeneration (*Witchley et al., 2013*). However, the roles for many of these genes with regionally restricted expression in muscle are poorly understood, and it is likely that many genes with regionally restricted expression in muscle and roles in positional information await identification.

Identification of muscle as a major site of expression of genes controlling regeneration and tissue turnover in adult planarians presented the opportunity for systematic characterization of positional information in an adult metazoan. To this end, we performed single-cell RNA sequencing on muscle cells isolated from 10 discrete regions along the planarian AP axis and found 44 genes for which expression within planarian muscle was restricted to specific AP domains. An RNA interference (RNAi) screen of many of these genes revealed two similar circuits each containing FGFRL and Wnt

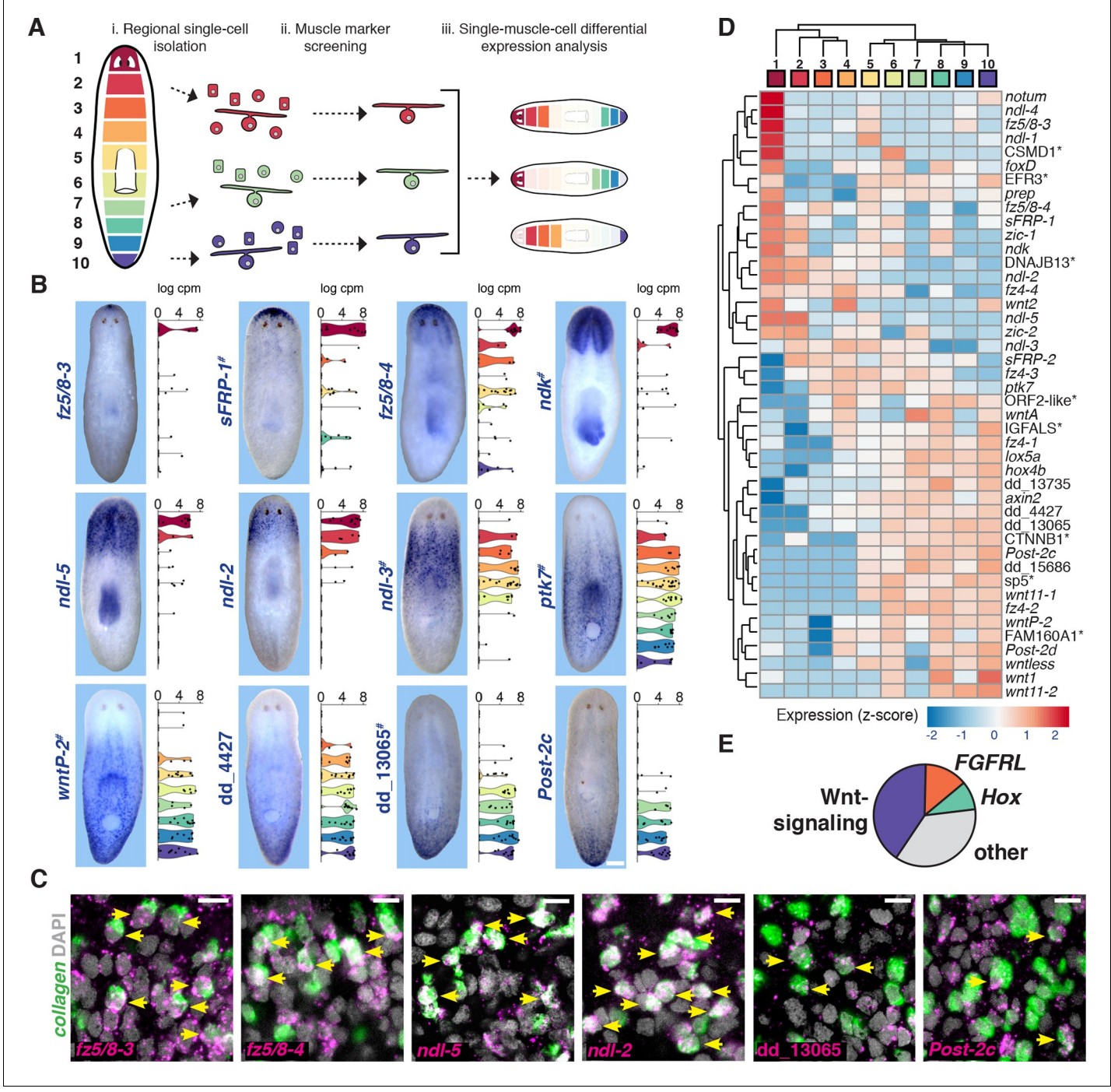

**Figure 1.** Single-muscle-cell RNA sequencing identifies regionally expressed genes on the planarian AP axis. (A) Single cells from each colored AP region were isolated by FACS and resultant cDNA was screened by qRT-PCR for muscle marker expression. Positive cells were sequenced and analyzed for differential expression. (B) Whole-mount in situ hybridization (ISH) (n=2 experiments) shows expression of a subset of new and previously known (#) muscle regionally expressed genes (mRGs). Images are representative of n>10 animals for new mRGs. Anterior, up. Scale bar, 100 µm. Right, violin plots show the expression distribution in muscle cells (black dots) within the 10 dissected regions. cpm, counts per million. (C) Double fluorescence ISH (FISH) show co-localization of several newly identified mRGs (magenta) and the muscle marker *collagen* (green). DAPI was used to label nuclei DNA (gray). Yellow arrows point to cells co-expressing both genes. Scale bar, 10 µm. (D) Heat map shows hierarchical clustering of the average expression per region of the 44 identified mRGs. Top color bar indicates dissected region. (*) marks genes that are named by best human BLASTx hits. (E) Pie chart shows the percentage, within the 44 genes shown in D, of Wnt-signaling genes, *FGFRL,* and *Hox* homologs.

The following figure supplements are available for figure 1:

*Figure 1 continued on next page*

*Figure 1 continued*

**Figure supplement 1.** Single-muscle-cell sequencing and analysis.
**Figure supplement 2.** 44 mRGs are distributed along the AP axis.

components that are required for the normal patterning of two distinct regions of the planarian body: the head and the trunk.

## Results

### Single muscle cell sequencing reveals 44 genes expressed in restricted domains along the AP axis

The prior identification of a single, body-wide cell type (body-wall muscle) expressing genes implicated in patterning in restricted domains (*Witchley et al., 2013*) raised the possibility that RNA sequencing of muscle cells could systematically identify components of this candidate adult positional information system. We sought such genes with regional expression in muscle utilizing single-cell RNA sequencing of muscle cells isolated from different regions along the AP axis. Non-dividing single cells from 10 consecutive regions along the AP axis (*Figure 1A*) were isolated by fluorescence activated cell sorting (FACS), and the resulting single-cell cDNA libraries were screened by qRT-PCR for expression of planarian muscle markers before sequencing (Methods, *Figure 1—figure supplement 1A–C*, *Supplementary file 1A*). Cells expressed an average of 3,253 transcripts, within the range reported for planarian single-cell libraries (*Wurtzel et al., 2015*). Principal component analysis (PCA) on the 177 single cells sequenced was performed using highly variable transcripts. Two significant principal components that separated cells by expression of muscle markers (PC1<0) and expression of neoblast markers (PC2<0) were identified (details in Methods, *Figure 1—figure supplement 1D–F*, *Supplementary file 1B*). PCA and *troponin* expression confirmed the identity of 115 muscle cells, and these 115 cells were used in all subsequent analyses (*Figure 1—figure supplement 1D*, *Supplementary file 1A*).

Single-cell differential expression (SCDE, [*Kharchenko et al., 2014*]) analysis of the data was used to identify regionally expressed genes in the muscle single cell sequencing data, because of its ability to identify transcripts of genes with known anterior and posterior expression patterns (details in Methods, *Figure 1—figure supplement 1G*). SCDE analyses of three different anterior-versus-posterior region comparisons (Materials and methods, *Figure 1A*, *Figure 1—figure supplement 1H*, *Supplementary file 1C–E*) identified transcripts of 99 genes as differentially expressed at p<0.005. To further validate the regional expression of these candidate genes, RNA probes were generated for all statistically significant transcripts (88/99 successfully amplified) and whole-mount in situ hybridization (ISH) was performed (*Figure 1B*). 18 genes with regional expression in muscle have been previously identified (*Figure 1—figure supplement 2*, [*Witchley et al., 2013*; *Vogg et al., 2014*; *Reuter et al., 2015*]). Although these three SCDE analyses correctly identified 13 of these 18 genes, those expressed in rare muscle cells (*wnt1*, *foxD*, *zic-1*), in shallow gradients (*sFRP-2*), or broadly (*wntA),* were below statistical significance (Materials and methods, *Figure 1—figure supplement 1H*, Figure 1—figure supplement 2B). Therefore, an additional 168 genes, for which transcripts showed non-significant differential expression in the SCDE analysis, were tested by ISH. ISH reveals expression in all tissue types, which might obscure detection of regional expression within muscle cells for some genes by this method. Nonetheless, ISH analysis verified 44 of these genes as regionally expressed (35/44 with p<0.005 in any of anterior-versus-posterior SCDE analyses) from the total 256 genes tested, including 26 previously not reported to be regionally expressed in muscle (*Figure 1B*, *Figure 1—figure supplement 2*, *Supplementary file 1F*). All newly identified regionally expressed genes tested were expressed at least in part in cells expressing the planarian muscle marker *collagen* (*Figure 1C*).

The term position control gene (PCG) has been used for genes with both regional adult expression, and patterning abnormal RNAi phenotypes or prediction by sequence to be in a pathway regulating planarian patterning (*Witchley et al., 2013*). The function for many such PCGs awaits

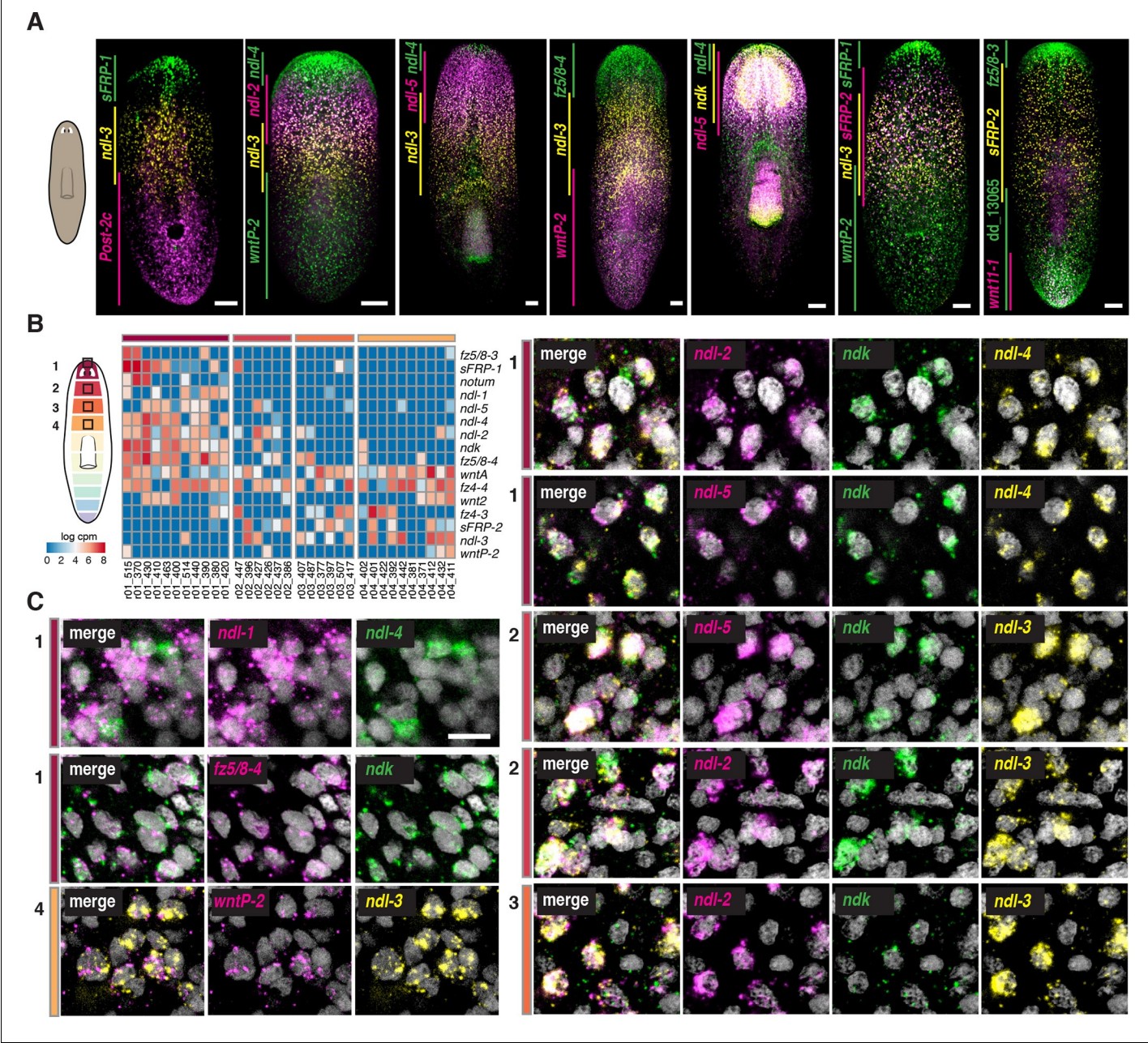

**Figure 2.** Co-expression of mRGs along the AP axis. (**A**) FISH using mRGs maps discrete domains of mRG expression onto the planarian AP axis. Bars on left indicate the approximate extent of the expression domain for each of the genes analyzed. Images are representative of n≥5 animals. Anterior, up. Scale bar, 100 μm. (**B**) Heatmap shows co-expression of anterior *FGFRL* and Wnt pathway mRGs in the four anterior regions indicated in the cartoon (1–4). Each column shows expression within a single cell with color bars above indicating the dissected region of origin for the cell. cpm, counts per million (**C**) FISH using different *FGFRL/ndl* probes and Wnt pathway mRGs show co-expression in the four regions depicted in the cartoon. Black boxes (1–4) in the cartoon in **B** show the region imaged for the FISH, as denoted by the number and colored rectangle next to the merged image. Scale bar, 10 μm. Images are representative of n≥5 animals.

The following figure supplements are available for figure 2:

**Figure supplement 1.** Axial mRG map and co-expression of multiple *FGFRL* genes and mRGs in the same muscle cell.

**Figure supplement 2.** Phylogenetic analysis of SMED-FGFRL proteins.

**Figure supplement 3.** Pattern of FGFRL/ndl family expression in *β-catenin-1* RNAi animals.

*Figure 2 continued on next page*

*Figure 2 continued*

**Figure supplement 4.** Inhibition of *FGFRL* genes does not significantly change expression of other members of the *FGFRL* family.

elucidation. Many of the genes identified here have as yet no known function and cannot be linked to known signaling pathways by sequence; we will therefore use in this manuscript the broadly inclusive term m̲uscle r̲egionally expressed g̲ene (mRG). Hierarchical clustering of the average expression per region of the 44 mRGs identified recapitulates the AP order of the regions (*Figure 1D*). Interestingly, the 44 identified mRGs identified here were comprised mainly of genes encoding Wnt-signaling components, Hox-family transcription factors, and fibroblast growth factor receptor-like (FGFRL) proteins (*Figure 1*E), suggesting that these gene families have prominent roles in providing positional information for maintaining and regenerating the planarian primary body axis.

## Regionally expressed genes in muscle, including *FGFRL* and Wnt-pathway genes, constitute an axial expression map in adult muscle

Combinatorial expression analysis using fluorescence ISH (FISH) of previously known mRGs and those newly described here generated a map depicting multiple, overlapping expression domains in planarian muscle along the planarian AP axis (*Figure 2A*, *Figure 2—figure supplement 1A*). Few genes, like *sFRP-2* and *ptk7* (*Gurley et al., 2010*; *Reuter et al., 2015*), were expressed broadly in the trunk. The posterior involves multiple overlapping expression domains of genes encoding Wnt, Hox, and novel proteins (*Petersen and Reddien, 2008*; *Adell et al., 2009*; *Iglesias et al., 2008*; *Reuter et al., 2015*; *Currie et al., 2016*). The anterior region involves overlapping expression domains of several components of the Wnt pathway and genes of the FGFRL family (*Petersen and Reddien, 2008*; *Rink et al., 2009*), some of which extended from the anterior head tip to varying posterior extents of the head and some were expressed in the pre-pharyngeal region (*Figure 2—figure supplement 2*).

FGFRL-family proteins, which lack the intracellular kinase domain present in FGFRs, have been little studied but are broadly conserved (*Figure 2—figure supplement 2*, [*Cebrià et al., 2002*; *Bertrand et al., 2009*]). The molecular mechanism of action of FGFRL proteins is not well understood. Planarians have six FGFRL-encoding genes, named *nou darake (ndk)*, the defining member of the *FGFRL* family (*Cebrià et al., 2002*), and *nou darake-like (ndl)-1* through *ndl-5* (*Figure 2—figure supplement 2*). All of these *FGFRL* genes were identified in the SCDE analyses as AP mRGs (*Figure 1D*). At least three of the five *FGFRL* genes with expression at the anterior-most region of the animal (*ndk, ndl-1, ndl-2, ndl-4, ndl-5*) were co-expressed in all 11 muscle cells isolated and sequenced from the anterior head tip (region 1); 10/11 of these cells also co-expressed *ndk* and *frizzled5/8–4 (fz5/8–4)* (*Figure 2B*). In the single muscle cells sequenced from the pre-pharyngeal region (region 4), 3/6 cells expressing *ndl-3* also expressed *wntP-2*. FISH also demonstrated co-expression of *FGFRL* and Wnt-pathway genes together in the same cells in regions where their expression domains overlapped (*Figure 2C*). Similarly, extensive co-expression of multiple mRGs in single muscle cells was observed in different regions along the entire AP axis (*Figure 2—figure supplement 1B*).

Wnt signaling is also required for maintenance of the AP axis, with β-*catenin-1* RNAi animals developing ectopic heads around the entire body during tissue turnover (*Petersen and Reddien, 2008*; *Gurley et al., 2008*; *Iglesias et al., 2008*). At early timepoints following RNAi (7–9 days after first RNAi feeding), β-*catenin-1* RNAi animals showed subtle posterior expansions of *ndl-5* and *ndl-2* expression domains (*Figure 2—figure supplement 3*). Later, ectopic expression of *ndl-5* in posterior and lateral locations occurred and preceded ectopic expression of *ndl-2* (9 days after first RNAi feeding) even before the appearance of ectopic eyes. Fully formed ectopic heads (21 days after first RNAi feeding) showed clear ectopic expression of the pre-pharyngeal mRG *ndl-3* (*Figure 2—figure supplement 3*). Gross anatomical changes in the AP axis are therefore accompanied by corresponding changes in *FGFRL* expression domains.

The axial expression map of genes in planarian adult muscle is reminiscent of regionalized gene expression patterns found during embryonic development in other species (*Pankratz and Jäckle,*

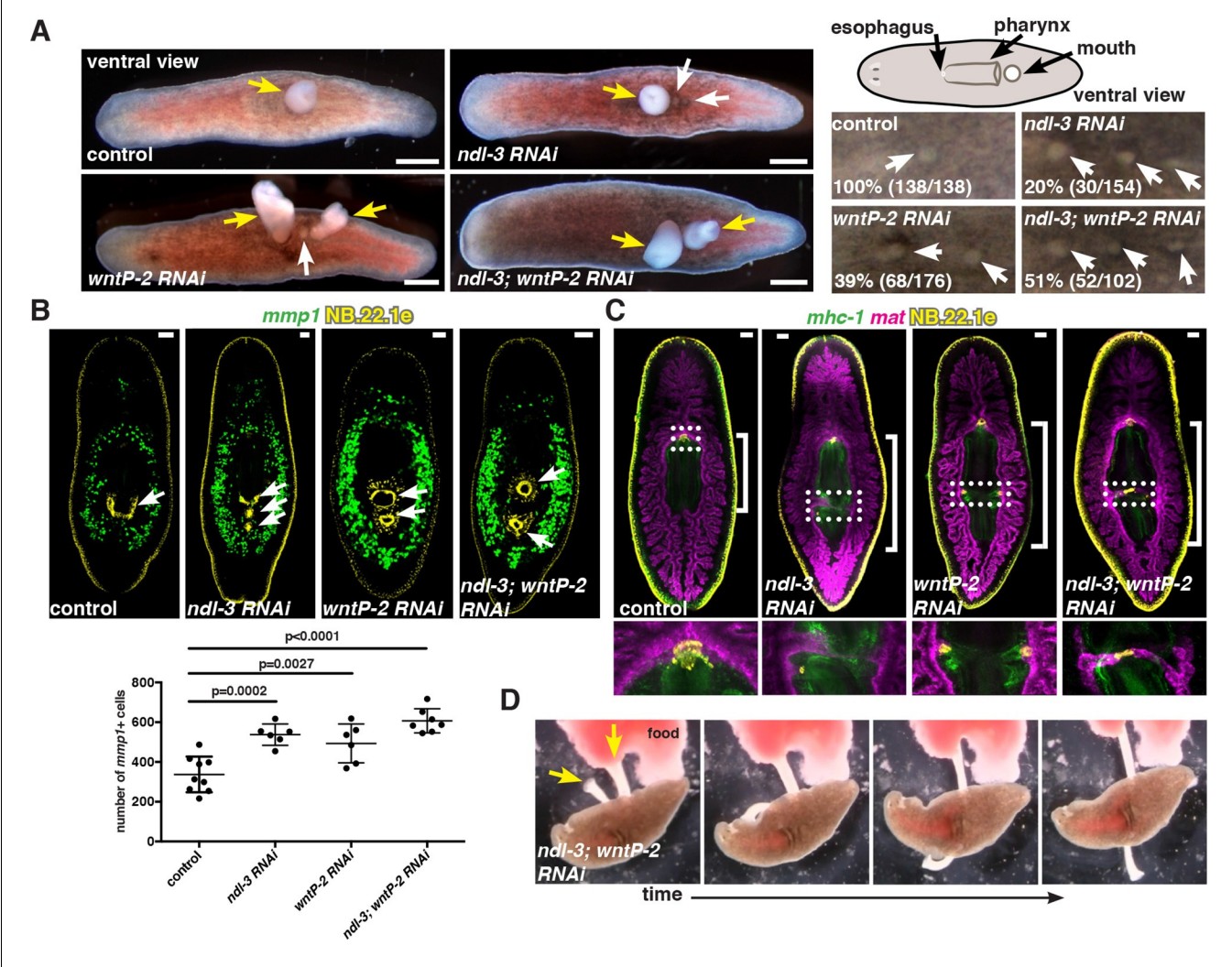

**Figure 3.** *ndl-3* and *wntP-2* restrict trunk positional identity. (A) Live, ventral images of ectopic pharynges and mouths in 20–30 day post-amputation (dpa) RNAi animals. Right top, cartoon depicts esophagus, pharynx, and mouth. Left, pharynges (yellow arrows) and ectopic mouths without a protruding pharynx (white arrows). Scale bar, 500 μm. Right bottom, mouths (white arrows) in 7 dpa RNAi animals. Anterior, left. Total number of animals were pooled from at least 2 independent RNAi experiments. (B) Increased numbers of para-pharyngeal *mmp1*+ cells in RNAi animals. NB.22.1e labels mouths. Graph below shows mean ± SD (n>8 animals/condition, 2 pooled experiments, One-way ANOVA). (C) Esophagus-gut connection in 20 dpa trunk fragments, region in dotted rectangle is shown at higher magnification below. FISH: *mat* (gut), *mhc-1* (pharynx), and NB.22.1e (esophagus). Bracket, pharyngeal cavity length. (D) Time-lapse images of an *ndl-3; wntP-2* RNAi animal eating liver through both pharynges (yellow arrows), see *Video 1*.

The following figure supplement is available for figure 3:

**Figure supplement 1.** *ndl-3* and *wntP-2* restrict the number of mouths and pharynges in the trunk region.

*1993*; *De Robertis et al., 2000*; *Jaeger et al., 2012*) and provides a tool to dissect adult positional identity maintenance and regeneration.

## Functional analysis of AP mRGs from major gene families

The regional expression of mRGs raises the possibility that many of these genes will have a role in controlling regional tissue identity. Therefore, we sought to determine with functional assays the roles of particular mRGs in the maintenance and/or regeneration of regional tissue identity. The prominence of a few gene families in the dataset of mRGs suggests that *FGFRL/Wnt/Hox* genes are

major patterning determinants of the planarian AP axis. We, therefore, performed extensive single and multi-gene RNAi to identify the roles of these genes in adult positional identity (*Supplementary file 1G*). Inhibition of single or combinations of *Hox* genes and a subset of *FGFRL* genes (*ndl-1, ndl-2, ndl-4,* and *ndl-5*) did not result in animals with a robust abnormal phenotype (*Supplementary file 1*G). Additionally, expression of other members of the *FGFRL* family was not affected under these RNAi conditions at the time-point analyzed (*Figure 2—figure supplement 4*). However, we found striking AP patterning phenotypes when inhibiting a subset of Wnt pathway components and *FGFRL* genes as described below.

### *ndl-3* and *wntP-2* restrict the number of mouths and pharynges in the planarian trunk

The *ndl-3* gene is expressed from below the eyes to the esophagus at the anterior end of the pharynx (Figure 1B, [*Rink et al., 2009*]), which is located centrally in the animal trunk (*Figure 3A*). *ndl-3* RNAi resulted in a striking phenotype: the formation of two or more mouths and two pharynges (*Figure 3A–C*, *Figure 3—figure supplement 1A–D*). The ectopic mouths and pharynges of *ndl-3* RNAi animals appeared within the trunk, posterior to the original mouth/pharynx location. This phenotype emerged both during tissue turnover in uninjured animals (*Figure 3—figure supplement 1C*) and following regeneration (*Figure 3A*, *Figure 3—figure supplement 1A*). In the case of regeneration, animals initially regenerated a single mouth/pharynx, but as regenerating animals grew following feeding, ectopic mouths and pharynges emerged. Inhibition of the posterior mRG *wntP-2/wnt11-5* also caused ectopic mouth and pharynx formation (*Figure 3A–C*, *Figure 3—figure supplement 1A–D*), in agreement with a recent report (*Sureda-Gómez et al., 2015*). Double RNAi of *ndl-3* and *wntP-2* was synergistic (*Figure 3A*, Fisher's exact test p<0.0001 for *ndl-3*, p=0.0153 for *wntP-2*, *Figure 3—figure supplement 1C*). Inhibition of *ndl-3* and *wntP-2* also resulted in pharyngeal cavity expansion (*Figure 3B,C*, *Figure 3—figure supplement 1B*), and in increased numbers of para-pharyngeal cells (*Figure 3B*), which express the matrix metalloproteinase *mmp1* (*Newmark et al., 2003*). In summary, when either *ndl-3* or *wntP-2* was inhibited, ectopic trunk structures were added sequentially as the animal grew and replaced tissues.

The planarian pharynx is a long muscular organ that can extend through the mouth to ingest food (*Reddien and Sánchez Alvarado, 2004*) and connects to the intestine through the esophagus at the medial anterior end of the pharyngeal cavity. In *ndl-3* and *wntP-2* RNAi animals, ectopic esophagi (NB.22.1e$^+$) formed in variable locations, including from the side of the pharyngeal cavity wall and from a gut branch crossing the pharyngeal cavity (*Figure 3C*). Despite variable positioning, ectopic pharynges always integrated through an esophagus into the intestine, demonstrating remarkable plasticity in the mechanisms underlying tissue organization. Ectopic pharynges in *ndl-3; wntP-2* RNAi or *wntP-2* RNAi animals were also functional – animals simultaneously projected both pharynges and each pharynx displayed independent food searching behavior, such as on opposite sides of the animal or in different directions (*Figure 3D*, *Video 1*).

### Inhibition of *ndl-3* and *wntP-2* affects the expression domains of trunk mRGs

Next, we examined whether trunk expansion in *ndl-3* and *wntP-2* RNAi animals with ectopic pharynges and mouths affected the mRG axial expression map described above. Anterior-most mRG expression domains (*sFRP-1* and *ndl-5*) were present and showed no overt changes following *ndl-3* and *wntP-2* RNAi (*Figure 4A–C*, *Figure 4—figure supplement 1A*). By contrast, the *ndl-3* expression domain was expanded to the ectopic posterior esophagus in *wntP-2* RNAi animals (*Figure 4A,D*). In both *wntP-2* and *ndl-3; wntP-2* RNAi animals, the expression domain of *sFRP-2* (*Figure 4—figure supplement 1D*) was also extended towards the animal posterior. By contrast, expression of the pre-pharyngeal mRGs

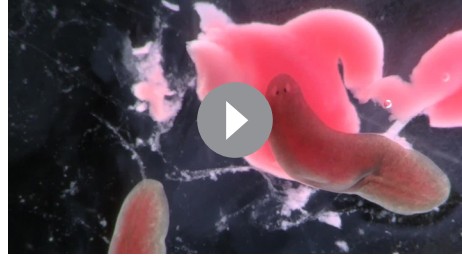

**Video 1.** Control, *wntP-2*, and *ndl-3; wntP-2* RNAi animals eating from one or two pharynges.

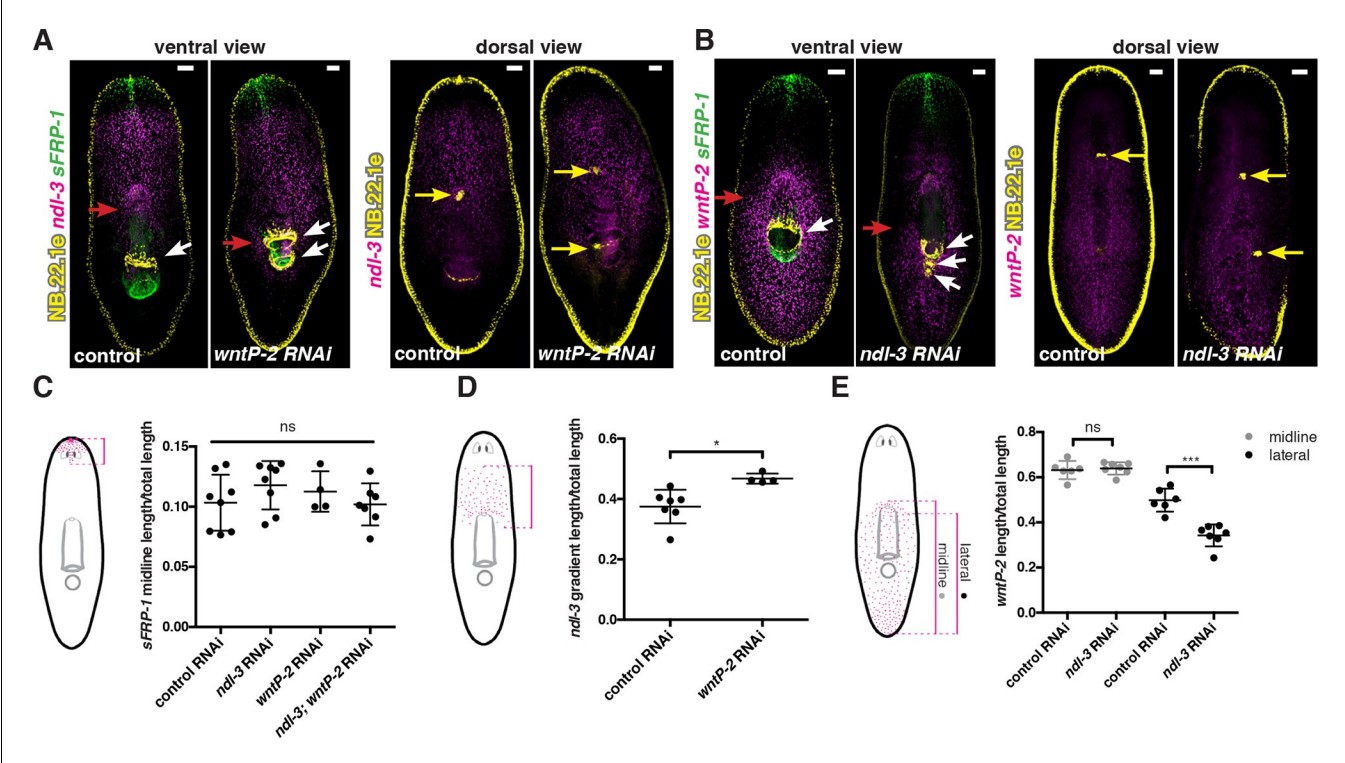

**Figure 4.** Trunk mRG gradients are shifted in *ndl-3* and *wntP-2* RNAi animals with ectopic pharynges/mouths. mRG expression analyses by FISH: (**A**) expanded expression of trunk mRG *ndl-3*, (**B**) reduction of the lateral expression of the posterior mRG *wntP-2*. Left panel, ventral view. Right panel, dorsal view. Red arrows point to the mRG expression domain boundary shifted. White arrows point to mouths. Yellow arrows indicate esophagus. Anterior, up. Scale bar, 100 μm. All FISH images are representative of n>8 animals per condition, and at least 2 independent RNAi experiments have been performed. (**C–E**) Graphs show quantification of the shifts in expression domains for the mRGs shown in the FISH experiments (mean ± SD, at least 3 independent experiments were pooled. One-way ANOVA for *sFRP-1*, unpaired Student's t-tests for *ndl-3* and *wntP-2*). Cartoons on the left depict the expression domain in the wild-type animal and the distance that was measured in each case. Length of expression domain measured was normalized by total length of the animal. All measurements were scored blind.

The following figure supplement is available for figure 4:

**Figure supplement 1.** *ndl-3* and *wntP-2* restrict trunk but not head or tail mRG expression domains in animals with ectopic pharynges/mouths.

*wnt2* and *ndl-2* was changed only slightly or not at all (*Figure 4—figure supplement 1B,C*). Conversely, the broad posterior expression domain of *wntP-2* was significantly reduced in *ndl-3* RNAi animals (*Figure 4B,E*). Expression of other posterior mRGs such as *fz4-1* and dd_13065 was still present in *ndl-3*, *wntP-2*, and *ndl3; wntP-2* RNAi animals (*Figure 4—figure supplement 1E,F*). Thus, both *ndl-3* and *wntP-2* are required for maintaining normal trunk tissue pattern including associated mRG expression domains, but not head or tail patterns of mRG expression. Altogether, these data suggest that the trunk patterning defects of *ndl-3* and *wntP-2* RNAi animals only affect local mRG expression within the axial map.

### *ndk*, *fz5/8–4*, and *wntA* control head patterning in planarians

In addition to trunk patterning phenotypes, we found that *fz5/8–4* RNAi caused ectopic eye formation and expansion of the brain posteriorly in both uninjured and regenerating animals (*Figure 5A,B*, *Figure 5—figure supplement 1A,C*). *fz5/8–4* showed graded anterior expression, strongest at the head tip, and brain expression (*Figure 1B*). The *fz5/8–4* RNAi phenotype was similar to that previously described for *ndk* and *wntA* RNAi (*Figure 5—figure supplement 1B*, [*Cebrià et al., 2002*; *Kobayashi et al., 2007*; *Adell et al., 2009*; *Hill and Petersen, 2015*]). *ndk* is expressed in head muscle and the brain (*Figure 1B*, [*Cebrià et al., 2002*; *Witchley et al., 2013*]) and restricts brain tissues to the animal head (*Cebrià et al., 2002*). *wntA* is expressed broadly, with strong expression at the

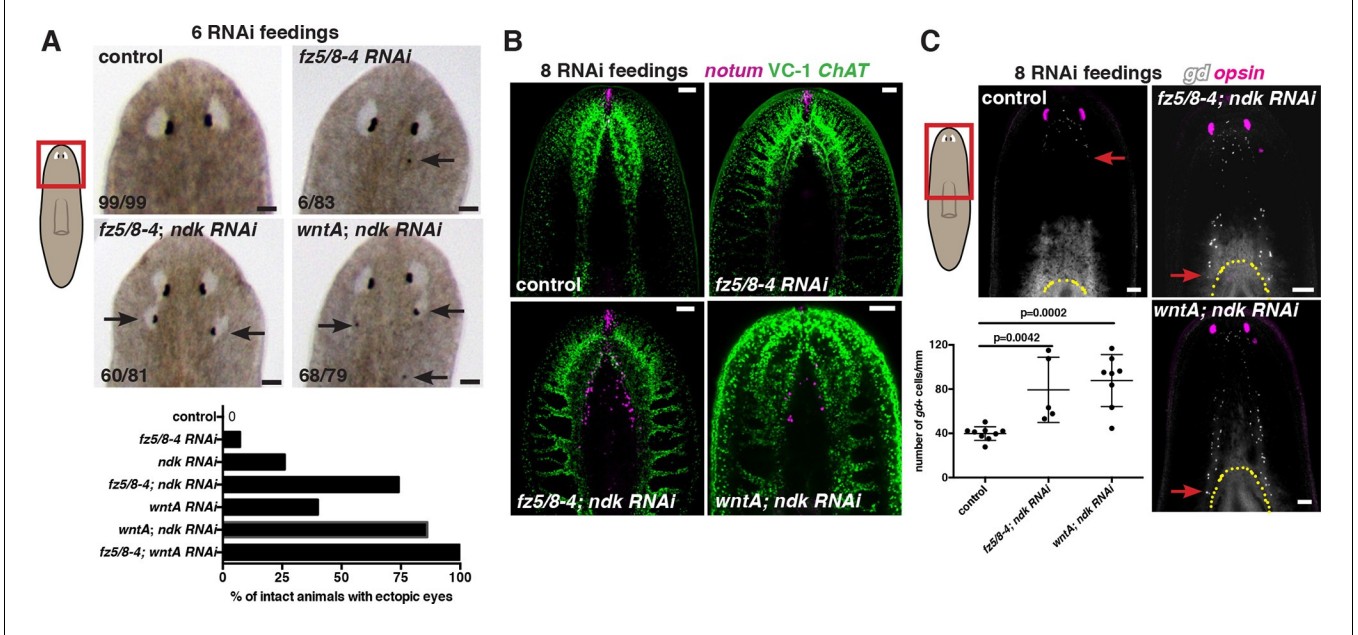

**Figure 5.** *fz5/8–4*, *wntA*, and *ndk* restrict head positional identity. (**A**) Posterior ectopic eyes seen in uninjured RNAi animals. Black arrows, ectopic eyes. Total number of animals have been pooled from 3 independent RNAi experiments. Cartoon on left shows area imaged. Graph below shows the percentage of intact animals with ectopic posterior eyes in each RNAi condition. (**B,C**) Posterior expansion of neuronal markers (**B**) *ChAT* and *notum* and eyes (anti-ARRESTIN/VC-1 antibody, images representative of n>5) and (**C**) *glutamic acid decarboxylase* (*gd*, red arrows mark posterior-most cell) and photoreceptor marker *opsin*. Cartoon on left shows area imaged. Below, graph shows increased $gd^+$ cell numbers, mean ± SD (n>5 animals/condition, 2 independent RNAi experiments, One-way ANOVA) normalized by the length from head tip to the esophagus.

The following figure supplement is available for figure 5:

**Figure supplement 1.** *fz5/8–4*, *wntA*, and *ndk* restrict the brain tissue to the head region.

posterior base of the brain (*Figure 1—figure supplement 2B*, [*Kobayashi et al., 2007*; *Adell et al., 2009*; *Hill and Petersen, 2015*]). *wntA; ndk* double RNAi animals showed a stronger phenotype in homeostasis (*Figure 5A,B*) and regeneration (*Kobayashi et al., 2007*), than did single gene RNAi animals. Double RNAi of *fz5/8–4* and either *ndk* or *wntA* also showed a synergistic effect during tissue turnover (*Figure 5A,B*, *Figure 5—figure supplement 1B*, Fisher's exact test p<0.0001 for both *ndk* and *wntA*). Additionally, RNAi of four out of five other *ndl* (*FGFRL*)-family members further enhanced the *fz5/8–4; ndk* double RNAi phenotype (*Figure 5—figure supplement 1D*, *Supplementary file 1G*), suggesting that multiple *FGFRL* genes synergize to control head pattern with *ndk*.

The *ndk* RNAi phenotype is poorly understood. For instance, are mRG expression domains expanded along with the brain in *ndk* RNAi animals, and what diversity of cell types expands posteriorly? Following eight RNAi feedings, *fz5/8–4; ndk*, *wntA; ndk*, and *fz5/8–4; wntA* double RNAi intact animals showed posterior expansion of multiple neuron classes from different head regions suggesting that the entire head-restricted nervous system expanded posteriorly (*Figure 5B,C*, *Figure 5—figure supplement 1E,F*). After eight RNAi feedings, a time point at which brain expansion was visible, non-neural *mag-1*⁺ adhesive gland cells (*Zayas et al., 2010*) showed normal distribution (*Figure 6—figure supplement 1A*). By 12 RNAi feedings, however, *mag-1*⁺ cells were disorganized (*Figure 6A*, *Figure 6—figure supplement 1D*), indicating that non-neural head cell types were eventually affected, but not visibly posteriorized, by these RNAi conditions.

We next examined whether the axial mRG map changed in RNAi animals with posterior ectopic eyes. Anterior-most (*sFRP-1* and *ndl-4*) and posterior (*wntP-2*, *wnt11-1*, and *wnt11-2*) mRG expression domains did not visibly change in *fz5/8–4; ndk* RNAi or *wntA; ndk* double RNAi animals, even under strong RNAi conditions (12 RNAi feedings, multiple ectopic eyes) (*Figure 6B,C*, *Figure 6—figure supplement 1B–D*). By contrast, *ndl-2* expression, which is normally restricted to a domain

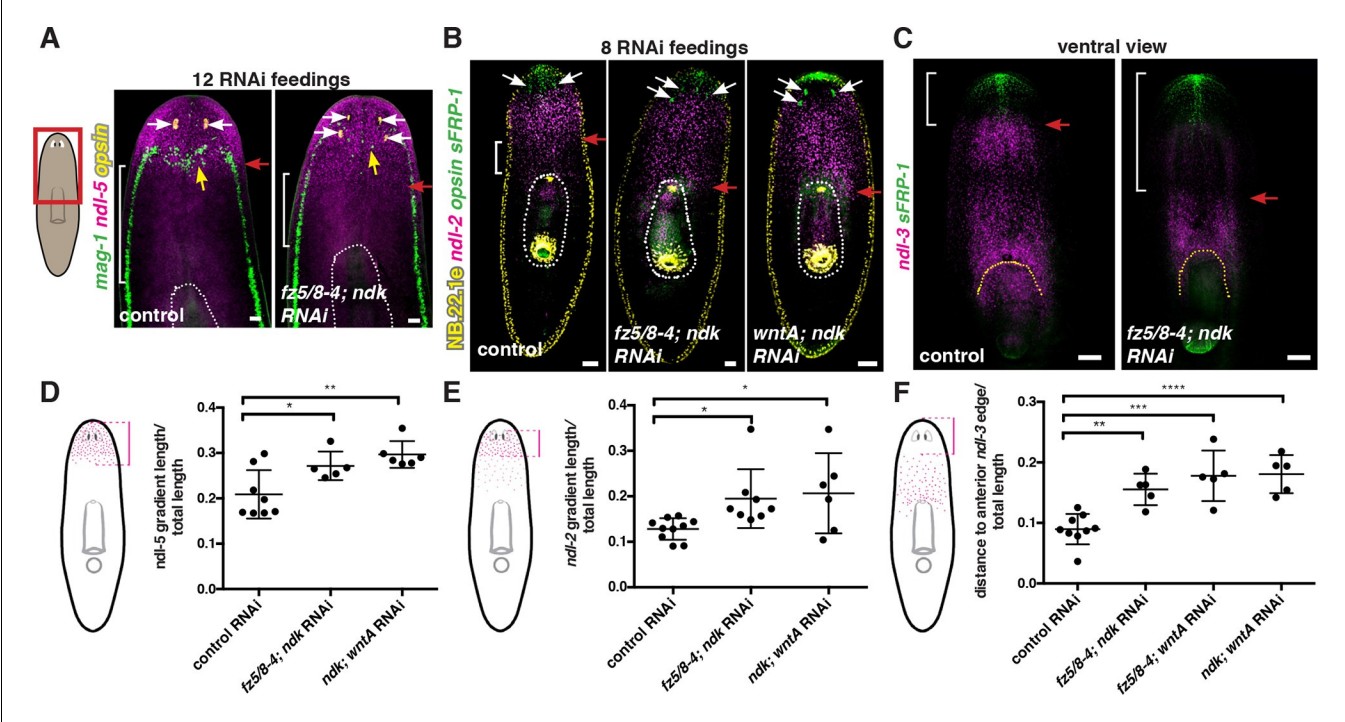

**Figure 6.** Anterior and prepharyngeal mRG gradients are shifted in *fz5/8–4, ndk*, and *wntA* RNAi animals with expanded brain tissue and ectopic eyes. (A,B) Expansion of mRG expression domains towards the animal posterior. White bracket marks distance between mRG posterior boundary and esophagus; red arrows mark expression domain shifts. White dotted lines outline pharynx. (A) *ndl-5*, and (B) *ndl-2.* (A) Disorganization of *mag-1* expression (yellow arrows). White arrows and *opsin* expression mark eyes. (C) Retraction of the pre-pharyngeal mRG *ndl-3*. Red arrows points to the shift towards the posterior of the anterior gradient boundary. White bracket indicates distance from the tip of the head to the anterior edge of the *ndl-3* gradient. In all panels, anterior is up. Scale bar, 100 μm. All FISH images are representative of n>10 animals and at least 2 independent RNAi experiments were performed. (D–F) Graphs show quantification of the expression domain shifts for the mRGs shown in the FISH experiments (mean ± SD, at least 3 independent experiments were pooled, One-way ANOVA). Cartoons on the left depict the expression domain in the wild-type animal and the distance (normalized to total length) that was measured in each case. All measurements were scored blind.

The following figure supplement is available for figure 6:

**Figure supplement 1.** *fz5/8–4, wntA*, and *ndk* locally restrict mRG expression in animals with expanded brain tissue and ectopic eyes.

immediately posterior to the eyes, was expanded into the pre-pharyngeal region after eight RNAi feedings (*Figure 6B,E*), and showed a more severe posterior expansion after 12 RNAi feedings (*Figure 6—figure supplement 1B*). Similarly, *ndl-5, ndk*, and *wnt2* expression domains extended posteriorly into the pre-pharyngeal region in RNAi animals with strong phenotypes (*Figure 6A,D*, *Figure 6—figure supplement 1C,E*). By contrast, the anterior end of the pre-pharyngeal *ndl-3* expression domain was significantly posterior-shifted (*Figure 6C,F*). Our results indicate that *ndk*, *wntA*, and *fz5/8–4* are required to restrict anterior tissues and associated expression domains of mRGs to the head region, while leaving the anterior tip, trunk, and tail mRG domains unaffected.

## Discussion

Single-cell sequencing has recently been used to identify transcriptomes for multiple planarian cell types (*Wurtzel et al., 2015*). We utilized single-cell sequencing to map axial gene expression within the planarian muscle. Planarian muscle was previously found to express several genes with known roles in adult tissue patterning in planarians, raising the possibility that muscle functions to produce a body-wide coordinate system of positional information (*Witchley et al., 2013*). Traditional RNA sequencing approaches to identifying candidate adult positional information in planarians is limited by the diversity of gene expression in heterogeneous tissue. The identification of a particular cell

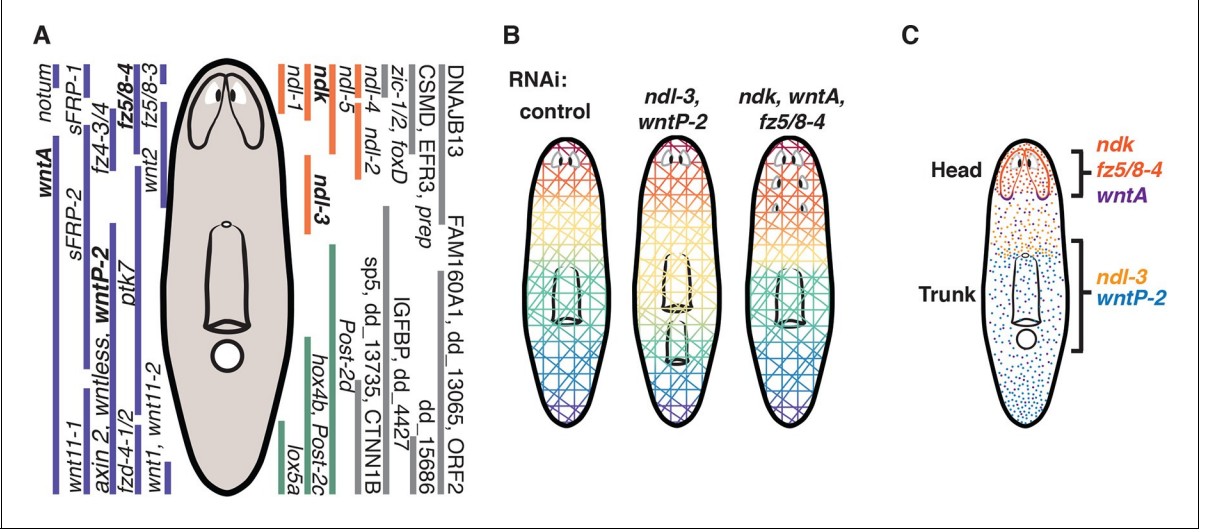

**Figure 7.** Two FGFRL-Wnt circuits control AP patterning in planarians. (**A**) Expression domains of all identified mRGs along the planarian AP axis. Wnt pathway (purple), *FGFRL* (orange), and *Hox* genes (green). In bold, genes shown here to be involved in maintaining regional identity. (**B**) Cartoons summarize the characterized RNAi phenotypes. *ndl-3* and *wntP-2* restrict the number of pharynges and mouths in the trunk region. *wntP-2* RNAi animals with ectopic pharynges/mouths have an expanded *ndl-3* domain whereas *ndl-3* RNAi animals with ectopic pharynges/mouths have a reduced *wntP-2* expression domain. *fz5/8–4, ndk*, and *wntA* restrict the brain tissue to the head. Inhibition of these genes results in ectopic posterior eyes, brain expansion, and expanded domains of head mRGs. (**C**) Expression domains of the two FGFRL-Wnt circuits are shown. Black brackets indicate the region controlled by the FGFRL-Wnt circuits.

type expressing genes associated with regional tissue identity allowed application of regional single-cell sequencing to surmount this challenge. We applied this approach to the AP axis, and identified mRGs that constitute an expression map of muscle cells of the planarian primary axis (*Figure 7A*). Coordinate systems of positional information, such as those proposed to control embryonic development (*De Robertis et al., 2000*; *Petersen and Reddien, 2009b*; *Niehrs, 2010*), might exist within adult tissues of many animals, including humans (*Rinn et al., 2006*), however there is little functional data regarding positional information and maintenance of the adult body plan. Here, we described several mRGs that work together to pattern and maintain two distinct body regions, the head and the trunk.

Constitutive regional expression of planarian orthologs to genes with key developmental roles in metazoans has been hypothesized to be important for multiple aspects of planarian body plan maintenance (*Reddien, 2011*). Here we show that both the planarian head and trunk require an FGFRL-Wnt circuit to maintain adult regional tissue identity. Different *FGFRL* and *Wnt* genes are used in the two body locations; however, in both cases, an *FGFRL* expression domain is juxtaposed by a posterior *Wnt* expression domain. Strikingly, inhibition of either gene (*FGFRL* or *Wnt*) caused posterior expansion and sequential duplications of structures normally found within the head and trunk regions, resulting in expanded brain and ectopic eyes in one case, and ectopic pharynges and mouths in the other (*Figure 7B*).

Following inhibition of any of the components of these FGFRL-Wnt circuits, the axial expression map shows local shifts in expression domains coincident with the expansion of specific regionally restricted tissues such as brain and pharynx (*Figure 7B*). These results suggest that muscle, a tissue found uniformly throughout the animal, marks different AP regions through combinatorial expression of mRGs. This implies that communication exists between muscle cells and the underlying region-specific tissues. Understanding the coordination between muscle cells and tissues within AP regions is necessary for determining how planarians are able to robustly maintain and regenerate their entire body plan. Positional information must be integrated into the decision to generate and pattern new tissue during planarian growth and regeneration. mRGs might therefore influence the regional behavior of neoblasts and/or their division progeny.

The two FGFRL-Wnt circuits described in this work are striking examples of body plan plasticity during homeostatic tissue turnover. In both cases, inhibition by RNAi of *FGFRL* genes (i.e., reduction or absence of the *FGFRL* expression domain) and inhibition of *Wnt* pathway components (i.e., expansion of the *FGFRL* expression domain) are coincident with the same phenotype of expanded regional identity. Future characterization of the biochemical properties of FGFRLs and elucidation of the mechanisms of interaction between Wnt/Fz pathways and FGFRLs might help in understanding this property. We propose that FGFRL proteins confine the regions where specific tissues in both the head and trunk can normally form and that the *Wnt* gene of each circuit acts by restricting tissues at the anterior end of its expression domain (*Figure 7B,C*). Given the similarities of the distinct FGFRL-Wnt circuits for patterning two different body regions, FGFRL-Wnt circuits might be broadly utilized, but presently underappreciated, patterning modules of animal body plans.

## Materials and methods

### Animals

Asexual *Schmidtea mediterranea* strain (CIW4) animals starved 7–14 days prior experiments were used.

### Muscle single cell isolation and library construction

Animals were dissected into 10 adjacent regions along the AP axis, and only the midline region (i.e., in between the ventral nerve cords) of each segment was utilized, to minimize heterogeneity caused by gradients expressed along the medio-lateral axes. 10 regions were chosen to balance consistency of amputation and AP resolution. The pharynx was dissected out and discarded for the regions 5 and 6. Fragments were dissociated into single cells. Single cell suspensions for each region were stained labeled with Hoechst, and non-dividing single cells were sorted by flow cytometry into 96 well plates containing 5 ul of total cell lysis buffer (Qiagen, Germany) with 1% β-mercaptoethanol. Subsequently, amplified cDNA libraries were made from each single cell using the SmartSeq2 method (*Picelli et al., 2013*; *2014*; *Wurtzel et al., 2015*), and tested by qRT-PCR for the expression of the muscle markers *collagen* and *troponin* (*collagen* Fw: GGTGTACTTGGAGACGTTGGTTTA, *collagen* Rv: GGTCTACCTTCTCTTCCTGGAAC; *troponin* Fw: ACAGGGCCTTGCAACTATTTTCATC, *troponin* Rv: GAAGCTCGACGTCGACAGGA). Cells expressing either or both of these muscle-specific genes (~5 in 96 cells) were used to make libraries using the Nextera XT kit (Illumina, Inc). Libraries were sequenced (Illumina Hi-seq) and fastq files generated by Illumina 1.5 and examined by fastqc.

### Identification of muscle cell expression profiles

Sequencing data was submitted to the GEO database as GSE74360. Each cell was sequenced twice, once with 80 bp reads and once with 40 bp reads, and reads from both sequencing runs were concatenated. Reads were trimmed using cutadapt to remove Nextera transposon sequences CTGTCTCTTATA and TATAAGAGACAG (overlap 11 bp) and low quality 3′ base pairs (quality score less than 30) before mapping to the dd_Smed_v4 assembly (http://planmine.mpi-cbg.de; [*Liu et al., 2013*]) using bowtie 1 (*Langmead et al., 2009*) with -best alignment parameter. Bowtie 1 was used because of its better sensitivity mapping <50 bp reads. Read counts from prominent mitochondrial and ribosomal RNAs (dd_smedV4_0_0_1, dd_Smed_v4_7_0_1, and dd_Smed_v4_4_1_1) were discarded. Reads from the same isotig were summed to generate raw read counts for each transcript (*Wurtzel et al., 2015*). Libraries with fewer than 1000 expressed (>2 reads) transcripts were discarded, leaving 177 cells, expressing an average of 3,253 unique transcripts with an average of 430,114 reads mapped (*Supplementary file 1A*, *Figure 1—figure supplement 1E*). Counts per million reads (cpm) were log transformed after addition of a pseudocount and used as expression values for violin plots and heatmap in *Figure 2B*. Principal component (PC) analysis on a set of highly expressed transcripts ($4 \leq$ mean expression $\leq 8$) with high variance (dispersion $\geq 1.2$) was extended to the entire set of transcripts to identify two significant PCs (Seurat [*Satija et al., 2015*], *Supplementary file 1B*). The transcripts defining PC1<0 are all found within the top 45 of the published set of muscle-enriched transcripts (*Wurtzel et al., 2015*). Two clear populations were separated by PC analysis, one of which included 115 cells that expressed *troponin* (>4 cpm) (*Figure 1—*

*figure supplement 1D,F*). These 115 muscle cells were used for all subsequent analysis (*Supplementary file 1A*). Average expression per region (Seurat) for each transcript (*Figure 1D*) or expression per cell (*Figure 2—figure supplement 1B*) was centered and scaled to generate expression z-scores used for heatmap visualization. Dendrograms show complete hierarchical clustering using Euclidean distance (*Figure 1D*, *Figure 2—figure supplement 1B*).

## Single cell differential analysis

Rv3.2.2 was used for all subsequent data analysis and visualization, relying on the following packages: Seurat (*Satija et al., 2015*), SCDE (*Kharchenko et al., 2014*), matrixStats, ROCR (*Sing et al., 2005*), ggplot2, RColorBrewer (http://colorbrewer2.org). To determine the best differential expression analysis method to identify putative mRGs, we tested three statistical methods: SCDE (*Kharchenko et al., 2014*), bimod (*McDavid et al., 2013*), and t-test (Student's t-test) for their ability to identify known mRGs. For each of the statistical tests, we compared cells from the head tip (region 1) versus those from the tail tip (region 10) and determined the rank and statistical significance of 10 canonical mRGs (*Figure 1—figure supplement 1G*). Based on its ability to identify mRGs present only in a subset of cells within a region (e.g., *wnt11-1, sFRP-1, notum*), we chose SCDE for all further differential expression analysis. Note that SCDE explicitly accounts for drop-out rates due to single-cell sequencing by calculating a probability distribution for each transcript in each cell before calculating differential expression between groups. To identify putative mRGs, we performed three differential expression analyses: anterior (regions 1, 2, 3; n=23 cells) versus posterior (regions 8, 9, 10; n=38 cells); head (region 1; 11 cells) versus post-pharyngeal (regions 7, 8, 9; n=35 cells); pre-pharyngeal (regions 2, 3, 4; n=22 cells) versus tail (region 10; n=12 cells) (*Figure 1A*). All transcripts with a |Z| score greater than 2.58 (p<0.005) in any of the SCDE analyses were screened by ISH (*Supplementary file 1C–E*). Z-scores corrected for multiple hypothesis testing are reported in *Supplementary file 1C–E*, however we used uncorrected Z-scores due to their ability to rank many more transcripts. In addition, 168 genes below our statistical cutoff were successfully amplified from cDNA and screened by ISH (*Supplementary file 1C–E*).

To determine if our method correctly classified transcripts as mRGs, we used a combined score from all three SCDE analyses (*Figure 1—figure supplement 1H*, [*Wan and Sun, 2012*]). If there is no differential expression of a gene along the AP axis, then the minimum p-value from any of the analyses, $p_{[1]} = min(p_1, \ldots, p_k)$ where $p_i$ is the p-value from $i^{th}$ analysis, is expected to follow a beta distribution with parameters 1 and k (*Tippett, 1931*). A differential expression score was calculated based on beta distribution, $S_{min} = -log(P(beta(1,k) \leq p_{[1]}))$, and used to rank all transcripts. Note that this scoring system only ranks transcripts, and that statistical significance of differential expression is only interpretable within the analysis performed. We then quantified how well our analyses, as scored by $S_{min}$, classified transcripts as mRGs, as determined by ISH validation. The receiver-operator curve plots the false positive rate versus the true positive rate for each value of $S_{min}$ based on ISH validation. The area under the curve (0.88, perfect classification = 1, random classification = 0.5) indicates that $S_{min}$ and therefore our SCDE analyses are able to correctly classify transcripts as mRGs visible by ISH.

## Gene cloning and whole-mount in situ hybridizations

Primers used to PCR amplify all planarian transcripts are listed in *Supplementary file 1C–E*. 44 mRGs were cloned from cDNA into the pGEM vector (Promega, Madison, WI). RNA probes were synthesized and nitroblue tetrazolium/5-bromo-4-chloro-3-indolyl phosphate (NBT/BCIP) colorimetric whole-mount in situ hybridizations (ISH) were performed as described (*Pearson et al., 2009*). Fluorescence in situ hybridizations (FISH) were performed as described (*King and Newmark, 2013*) with minor modifications. Briefly, animals were killed in 5% NAC before fixation in 4% formaldehyde. Following treatment with proteinase K (2 µg/ml) and overnight hybridizations, samples were washed twice in pre-hyb buffer, 1:1 pre-hyb:2X SSC, 2X SSC, 0.2X SSC, PBST. Subsequently, blocking was performed in 5% casein (10X solution, Sigma, St. Louis, MO) and 5% inactivated horse serum PBST solution when anti-DIG or anti-DNP antibodies were used, and in 10% casein PBST solution when an anti-FITC antibody was used. Post-antibody binding washes and tyramide development were performed as described (*King and Newmark, 2013*). Peroxidase inactivation with 1% sodium azide was done for 90 min at RT. Live animal images were taken with a Zeiss Discovery Microscope.

Fluorescent images were taken with a Zeiss LSM700 Confocal Microscope. Co-localization analysis of FISH signals was performed using Fiji/ImageJ. For each channel, histograms of fluorescence intensity were used to determine the cut-off between signal and background. All FISH images shown are maximal intensity projections. A median filter was applied using the ImageJ Despeckle function. Images are representative of results seen in >5 animals per panel.

## RNAi

dsRNA was prepared from in vitro transcription reactions (Promega) using PCR-generated templates with flanking T7 promoters, followed by ethanol precipitation, and annealed after resuspension in water. dsRNA was then mixed with planarian food (liver) (*Rouhana et al., 2013*) and 2 ul per animal of the liver containing dsRNA was used in feedings. For the RNAi screen (*Supplementary file 1G*), the following feeding protocol was used: animals were fed six times in three weeks, cut in four pieces (head, pre-pharyngeal, trunk and tail pieces), allowed to regenerate for 10 days, fed all together six times in another three weeks, and cut again into the four pieces described above. Seven days following amputation (7 dpa), trunk pieces were scored (*Supplementary file 1G*, *Figure 3A* right panels, *Figure 3—figure supplement 1A*, and *Figure 5—figure supplement 1A*), and fixed at day 20 following amputation for further analysis (*Figure 3* and *Figure 4*). Five to 10 trunk pieces were kept after the first regeneration cycle, and were fed once a week for another 12 weeks and were scored after that period (homeostasis, *Supplementary file 1G*). Animals for homeostasis RNAi experiments for trunk patterning studies (*Figure 3—figure supplement 1C*) were fed eight times in four weeks and scored a week after the last feeding. In RNAi experiments for head patterning analysis, animals were fed eight times in four weeks, scored after the first six feedings (*Figure 5A*, *Figure 5—figure supplement 1B,D*), and fixed seven days after the last feeding (without amputation). For longer time point experiments, animals were fed twelve times in six weeks and fixed seven days after the last feeding. For all RNAi conditions tested, the total amount of dsRNA per feeding per animal was kept constant. Therefore, for example, when RNAi of two or more genes was performed, dsRNA for each gene was diluted in half. For combinations of dsRNAs, synergistic effects of double RNAi were calculated using Fisher's exact test (*Figure 3A*, *5A*, *Figure 3—figure supplement 1C*). Minimum sample sizes were estimated using difference of proportion power calculation with h=0.4 (ectopic mouths) or h=0.8 (ectopic eyes), sig.level=0.05, and power=0.8 (n=98.1 or n=24.5). RNAi animals with ectopic pharynges/mouths were treated with 0.2% chlorotone, which results in muscle relaxation and pharynx protrusion through the mouth (*Figure 3A*, left panels). For RNAi enhancement experiments of *ndl-3; wntP-2* RNAi with β-*catenin-1*, animals were fed five times with a combination of *ndl-3* and *wntP-2*, and starting in the sixth feeding, β-*catenin-1* or control dsRNA was added to the mix of *ndl-3* and *wntP-2* for another three, four, or five feedings (being a total of eight, nine or 10 feedings, *Figure 3—figure supplement 1E*). For the β-*catenin-1* RNAi experiment shown in *Figure 2—figure supplement 3*, animals were fed once, twice, or four times with β-*catenin-1* or control dsRNA. Animals were fixed at different days after the first RNAi feeding.

## Quantitative reverse-transcriptase PCR

Samples were processed and analyzed as described (*Owen et al., 2015*). Briefly, total RNA was isolated from fragments from individual intact worms or from individual regenerated fragments, as indicated by cartoons in figures, in 0.75mL Trizol (Life Technologies, Carlsbad, CA) following manufacturer's instructions. Samples were homogenized for 30s using TissueLyser II (Qiagen). Following RNA purification and resuspension in dH20, concentrations for each sample were measured by Qubit using RNA HS Assay Kit (Life Technologies). 5 ng of RNA were treated with 1U amplification-grade DNAse I (Life Technologies) for 15 min at room temperature before DNAse heat-inactivation for 10 min at 65°C in the presence of 2.5 mM EDTA. Multiplex reverse-transcription and 15 cycles of PCR amplification were performed on DNAse-treated RNA using pooled outer primers at 50 nM each and Superscript III/Platinum Taq enzyme mix (*Supplementary file 1H*). Following outer primer digestion with ExoI (15U, New England Biolabs, Ipswich, MA), samples were diluted to 500pg/ul and checked for presence of *g6pd* by qRT-PCR (7500 Fast PCR System, Applied Biosystems). Samples and inner primers were loaded onto a 96.96 Dynamic Array Fast IFC chip (Fluidigm BioMark) and analyzed as described (*Supplementary file 1H*, [*Owen et al., 2015*]). Ct values from two technical replicates were averaged and normalized by the average Ct value of three

housekeeping genes (*g6pd, clathrin,* and *ubiquilin,* [*van Wolfswinkel et al., 2014*]) to generate ΔCt values. Log$_2$ fold-changes were determined by the ΔΔCt method by calculating the difference from the average ΔCt value of control RNAi replicates. Heatmap of average ΔΔCt values was generated by pheatmap in R. Bar graphs show mean ΔΔCt +/- standard deviation with individual ΔΔCt values. Statistical tests (unpaired Student's t-test or one-way ANOVA followed by Dunnett's multiple comparisons test) were performed between individual ΔΔCt values.

## Expression domain quantification

FISH of the mRG of interest was performed in control RNAi animals and RNAi animals showing phenotypes (ectopic pharynx or ectopic eyes) in at least three independent experiments, and images taken with same intensity settings within an experiment. The extent of an mRG expression domain was measured in ImageJ as depicted in cartoons by blind scoring maximal intensity projections. For *ndl-2,* the extent of the domain with strong expression and not total expression was measured. The length of the mRG expression domain was normalized by the length of the animal. Statistical analysis of expression domain shifts were determined by one-way ANOVA followed by Dunnett's multiple comparisons test.

## Immunostainings

Animals were fixed as for in situ hybridizations and then treated as described (*Newmark and Sánchez Alvarado, 2000*). A mouse anti-ARRESTIN antibody (kindly provided by Kiyokazu Agata) was used in a 1:5000 dilution, and an anti-mouse-Alexa conjugated antibody was used in a 1:500 dilution.

## Cell quantification and statistical analysis

Numbers of *cintillo$^+$* and *gd$^+$* cells were counted and normalized by the length between the anterior tip of the animal and the esophagus in control, *fz5/8–4; ndk,* and *wntA; ndk RNAi* animals after eight RNAi feedings (*Figure 5C*, *Figure 5—figure supplement 1E*). One-way ANOVA and Dunnet's posttest were used to determine significant differences between the different conditions and the control. Similarly, cells expressing the metalloproteinase *mmp1* were counted in control, *wntP-2, ndl-3,* and *ndl-3; wntP-2* RNAi trunk pieces after 12 RNAi feedings and two rounds of regeneration (20 dpa, screen RNAi protocol, *Figure 3B*). One-way ANOVA and Dunnet post-test were used to determine significant differences between the different conditions and the control. Minimum sample-size estimations were calculated using balanced one-way analysis of variance power calculation with k=4 (*mmp1*) or k=3 (*cintillo* and *gd*), f=0.8, sig.level=0.05, and power=0.8 (n=5.3 or n=6.1).

## Phylogenetic analysis: accession numbers

Genbank: *Homo sapiens*: NP_068742.2 (FGFRL1). *Mus musculus*: NP_473412.1 (FGFRL1). *Xenopus tropicalis* NP_001011189.1 (FGFRL1). *Strongylocentrotus purpuratus* NP_001165523.1 (FGFRL1). Dj, *Dugesia japonica*: BAC20953.1 (Ndk), BAP15931.1 (Ndl-2), BAQ21471.1 (Ndl-3), BAQ21471.1 (Ndl-1). *Nematostella vectensis* XP_001635234.1 (FGFRL-1). Uniprot: *Ciona intestinalis* F7BEX9 (FGFRL1). Genomic database: *Capitella sp. I*: CAPC1_170033 (FGFRL1). *Lottia gigantean*: LOTGI_167118 (FGFRL1). *Schistosoma mansoni*: Smp_052290 (Ndk), Smp_036020 (Ndl-5) Planmine/Genbank: *Schmidtea mediterranea*: dd_11285/ADD84674.1(Ndk), dd_12674/ADD84675.1 (Ndl-4), dd_5102/ AFJ24803.1(Ndl-5), dd_6604/ADD84676.1 (Ndl-3), dd_8310 (Ndl-1), dd_8340 (Ndl-2).*Dendrocoelum lacteum*: Dlac_193209/JAA92597.1 (Ndk), Dlac_194186/JAA92596.1 (Ndl-4), Dlac_184398 (Ndl-5), Dlac_178408 (Ndl-3-2), Dlac_182339 (Ndl-3-1), Dlac_189993 (Ndl-1), Dlac_170672 (Ndl-2-1), Dlac_181923/JAA92595.1 (Ndl-2-2). *Planaria torva*: Ptor_24279 (Ndk-1), Ptor_18635 (Ndk-2), Ptor_24521(Ndl-4-1), Ptor_34251 (Ndl-4-2), Ptor_36400 (Ndl-5-1), Ptor_68870 (Ndl-5-2), Ptor_24828 (Ndl-3-1), Ptor_27132 (Ndl-3-2), Ptor_29905 (Ndl-1), Ptor_23702 (Ndl-2). *Polycelis tenuis*: Pten_63627 (Ndk-1), Pten_14428 (Ndk-2), Pten_6171 (Ndl-4), Pten_43037 (Ndl-5), Pten_46975 (Ndl-3), Pten_39799 (Ndl-1-1), Pten_47685 (Ndl-1-2), Pten_41107 (Ndl-2). *Polycelis nigra*: Pnig_15421 (Ndk-1), Pnig_6593 (Ndk-2), Pnig_29523 (Ndl-4-2), Pnig_3947 (Ndl-4-1), Pnig_25001 (Ndl-5), Pnig_3933 (Ndl-3), Pnig_25308 (Ndl-1), Pnig_22111 (Ndl-2).

## Acknowledgements

We thank Inma Barrasa for help with RNA-seq analysis and Dayan Li for designing collagen and cadherin primers. We are grateful to the work of Aneesha Tewari and Jennifer Cloutier in designing and verifying qPCR primers.

## Additional information

### Funding

| Funder | Author |
| --- | --- |
| National Institute of General Medical Sciences | Peter W Reddien |
| Howard Hughes Medical Institute | Peter W Reddien |

The funders had no role in study design, data collection and interpretation, or the decision to submit the work for publication.

### Author contributions

MLS, LEC, PWR, Conception and design, Acquisition of data, Analysis and interpretation of data, Drafting or revising the article; TR, Acquisition of data

### Author ORCIDs

Lauren E Cote, http://orcid.org/0000-0002-1772-7447
Peter W Reddien, http://orcid.org/0000-0002-5569-333X

## Additional files

### Supplementary files

• Supplementary file 1. Summary of mRG data from single-cell RNA sequencing data, ISH screen, and RNAi screen. (A) Single-cell library information. (B) Top 12 significant genes defining principal components (PC) 1 and 2. (C) Anterior (regions 1-3) versus Posterior (regions 8-10) SCDE analysis and ISH results. (D) Head (region 1) versus post-pharyngeal (regions 7-9) SCDE analysis and ISH results. (E) Pre-pharyngeal (regions 2-4) versus post-pharyngeal (region 10) SCDE analysis and ISH results. (F) List of 44 mRGs. (G) RNAi summary. (H) qRT-PCR primers.

### Major datasets

The following dataset was generated:

| Author(s) | Year | Dataset title | Dataset URL | Database, license, and accessibility information |
| --- | --- | --- | --- | --- |
| Scimone ML, Cote LE, Rogers T, Reddien PW | 2015 | Single-muscle-cell transcriptome profiling along the planarian AP axis | http://www.ncbi.nlm.nih.gov/geo/query/acc.cgi?acc=GSE74360 | Publicly available at the NCBI Gene Expression Omnibus (Accession no: GSE74360). |

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
