## [Decision Letter]

Thank you for submitting your work entitled "Two FGFRL-Wnt circuits organize the planarian AP axis" for consideration by *eLife*. Your article has been reviewed by three peer reviewers, and the evaluation has been overseen by Alejandro Sánchez Alvarado as Reviewing Editor and Fiona Watt as the Senior Editor. One of the three reviewers has agreed to reveal her identity: Yukiko Yamashita.

The reviewers have discussed the reviews with one another and the Reviewing Editor has drafted this decision to help you prepare a revised submission.

Summary:

In the manuscript by Scimone, et al., the authors build upon their previous study of spatial signaling pathways being transcribed by muscle nuclei: Witchley et al., 2013. In order to describe the heterogeneity of signals transcribed by muscles, and how those signals may change along the axis of the worm, the authors use single cell RNA deep sequencing of 115 muscle cells isolated from different axial levels. This manuscript identifies genes that are expressed regionally along the anterior/posterior axis in planarian body wall muscle, which are thought to provide a coordinate system for axial information. Trying several analysis methods, the authors settle on one and then perform an in situ hybridization screen of 252 transcripts to look for regionally restricted transcripts. The RNA sequencing of single muscle cells identified 44 new regionally-expressed genes, many of which are members of the Wnt, FGFRL, and Hox gene families. Knockdown of these genes resulted in two overall classes of phenotypes: those that affect trunk identity, with formation of ectopic mouths, and those that affect anterior identity, with formation of ectopic photoreceptors. In both cases, the genes required for head or trunk identity consist of one Wnt and one FGFRL, suggesting that these proteins may form a regulatory circuit conveying regional axial information to underlying tissues in the animal.

Essential revisions:

The use of single-cell RNA-sequencing to identify transcripts expressed in particular cell types is a novel, powerful technique in planarians. The finding that signaling pathways such as Wnt and FGFRL cooperate to establish axial patterning is not altogether surprising, but understanding how this coordinate system functions is valuable. However, before publication some changes need to be made to improve the clarity of the story.

1) This manuscript should be expanded into a full article due to the vast amounts of data currently buried in supplemental (Figure 1—figure supplement 3 for example). Additionally, a very detailed protocol and sequence analysis accompanying this article would be helpful. Currently, sequencing depth or testing of sensitivity by mRNA spike-ins is not discussed or performed, and should be strongly considered to expand upon.

2) The selection criteria for these 44 new regionally-expressed genes is not clear from the text. Additionally, in screening 252 genes by in situ hybridization, why were 166 non-statistically-significant genes in this analysis? Also, in Figure 1—figure supplement 2, why are only 38 in situs instead of 44? Stating the reasons for these inclusions and omissions will improve the clarity of the manuscript.

3) The claim that *WntP-2* and *ndl-3* only affect trunk identity and not head identity is poorly supported by the data as shown. The SFRP-1 expression patterns in Figure 2 seem to be moderately expanded, and it is difficult to tell how the authors determine the boundaries of these domains. Can they be quantified to show reproducibility across a population instead of in a single image? Similarly, the expression pattern boundaries shown in Figure 2—figure supplement 2 are difficult to see, and quantification would help.

4) The authors suggest in the conclusion that the Wnt signals form the posterior boundary for the FGFRLs. To prove that this system functions this way, does knockdown of multiple Wnts alter the FGFRL distribution? Does overall perturbation of global Wnt signaling (by knockdown of β-catenin or APC) alter patterning of these genes?

5) Levels of *WNTP-2, ndl-3, fz5/8-4, ndk* RNAi efficacy should be quantified by qRTPCR or RNAseq, as is standard in the planarian field.

6) Because the phenotype of *wntA, ndk*, and *fz5/8-4* are all in a similar direction (head expansion), there cannot be a negative arrow from *wntA* to *ndk* in the model in Figure 4. Please modify accordingly if you propose this is a genetic arrow, or increase resolution if you propose this is a small-scale boundary restriction.

7) The authors did not put the ndl's in the context of regional head/tail repression by βcat/APC. In the case of the trunk domain expanding posterior, can this be suppressed by βcat RNAi's anteriorizing effects?

8) In the end, this is largely an ndl story as well as trying to get a handle on what the muscle cells are capable of in terms of signaling complexity. The authors state that 4/5 ndl's were differentially detected in muscle cells, yet it is hard to get a feeling for how they are regionally restricted, how they overlap with each other, and what how each of their phenotypes affect patterning as a whole. The authors should show the RNAi phenotypes for each of the ndl's and show the effects on the patterning of the others as well as the WNT's used in this manuscript. Secondly, they should quantify the overlap in each of the genes and support this by co-detection in single muscle cells using the data already in hand. The authors missed an opportunity here to perform some analyses on what mRGs were detected in single muscle cells – they should perform such an analysis (instead of the average expression for a region shown in Figure 1). For example, do all ndk-expressing muscles also express *wntA* and/or *fz5/8-4*? Are there transcription factor differences that can explain the expression profile of a given muscle? This would be very informative.

---

## [Author Response]

*Essential revisions: The use of single-cell RNA-sequencing to identify transcripts expressed in particular cell types is a novel, powerful technique in planarians. The finding that signaling pathways such as Wnt and FGFRL cooperate to establish axial patterning is not altogether surprising, but understanding how this coordinate system functions is valuable. However, before publication some changes need to be made to improve the clarity of the story. 1) This manuscript should be expanded into a full article due to the vast amounts of data currently buried in supplemental (Figure 1—figure supplement 3 for example). Additionally, a very detailed protocol and sequence analysis accompanying this article would be helpful. Currently, sequencing depth or testing of sensitivity by mRNA spike-ins is not discussed or performed, and should be strongly considered to expand upon.*

We agreed with the reviewers and have expanded this manuscript into a full article. We also agree with the reviewers that more detail for the single-cell sequencing could enhance the paper. Therefore we have added details of the sequencing analysis to the Results, including average number of transcripts detected per cell and to the Materials and methods, including average number of reads per cell, with additional references to [Supplementary-material SD1-data] where the values for the individual cells are listed. Of note, a recent manuscript from our lab describes the single cell sequencing protocol used here in detail as well, and we also have added reference to this published detailed protocol to the paper (Wurtzel et al., 2015). We have included that the genes defining the final muscle cells used for the differential expression analysis were within the muscle-enriched set found by Wurtzel et al., 2015. The Smart-seq2 library preparation protocol (Picelli et al., Nature Methods, 2013), which we used for this work, does not use EERC spike-ins for normalization. Single-cell sequencing does suffer from drop-out rates, which is accounted for by the statistical modeling present in SCDE (Kharchenko et al., 2014), and this is noted in the Materials and methods.

*2) The selection criteria for these 44 new regionally-expressed genes is not clear from the text. Additionally, in screening 252 genes by in situ hybridization, why were 166 non-statistically-significant genes in this analysis? Also, in Figure 1—figure supplement 2, why are only 38 in situs instead of 44? Stating the reasons for these inclusions and omissions will improve the clarity of the manuscript.*

We thank the reviewers for pointing out this area of confusion. 35 of the 44 mRGs described were within the 99 genes that had statistically significant p-values from the SCDE analyses performed and that also showed regional expression by in situ hybridization. The additional 9 mRGs verified by in situ hybridization came from 168 genes below the significant p-value in the SCDE analyses. The reason for including these 168 non-significant genes was because of the absence of a few previously known mRGs within the 99 (p-value<0.005) group. mRGs that were expressed in shallow gradients (e.g., *sFRP-2*) or in very few cells (*wnt1, foxD, zic-1*) were not present within the 99 significant genes. We therefore sought to examine the possibility that the threshold of p-value<0.005 was too stringent to capture all mRGs in the data. On the other hand, lowering the threshold would increase the false positive rate. We therefore tested additional genes with different significance values to seek additional mRGs. Because only 4 more mRGs (plus 5 previously known mRGs) were found with this approach, we concluded we were near saturation of the data. i.e., further sampling below the significance level threshold set would reveal mainly false positives not validated by ISH (see Figure 1—figure supplement 1, Methods subsection “Single cell differential analysis”). We re-wrote this section in the Results (subsection “Single muscle cell sequencing reveals 44 genes expressed in restricted domains along the AP axis”) to make this approach clearer.

Previously, we did not include all 44 genes in Figure 1—figure-supplement 2 because some of those genes were already shown in Figure 1. Now, for clarity purposes, we include all 44 genes in Figure 1—figure supplement 2.

*3) The claim that WntP-2 and ndl-3 only affect trunk identity and not head identity is poorly supported by the data as shown. The SFRP-1 expression patterns in Figure 2 seem to be moderately expanded, and it is difficult to tell how the authors determine the boundaries of these domains. Can they be quantified to show reproducibility across a population instead of in a single image? Similarly, the expression pattern boundaries shown in Figure 2—figure supplement 2 are difficult to see, and quantification would help.*

We agreed with the reviewers about the challenge in visualizing shifts in individual images of expression domains from a figure. Therefore, we now added quantifications of all the expression domains that we thought might be shifting between the different RNAi conditions. We now demonstrate statistical significant shifts for the expression domains of the trunk mRGs *ndl-3, wntP-2*, and *sFRP-2* in *wntP-2 / ndl-3* RNAi conditions (Figure 4, Figure 4—figure supplement 1). ndl-2, showed subtle-to-no changes in its expression domain between conditions, and we describe it accordingly. The particular case of *sFRP-1* asked for showed no significant changes in its expression domain between the control and the RNAi groups. We also similarly quantified anterior expression domains, now shown in Figure 6.

*4) The authors suggest in the conclusion that the Wnt signals form the posterior boundary for the FGFRLs. To prove that this system functions this way, does knockdown of multiple Wnts alter the FGFRL distribution? Does overall perturbation of global Wnt signaling (by knockdown of β-catenin or APC) alter patterning of these genes?*

We re-wrote those conclusions to be more specific about the role of Wnt signaling setting in the FGFRL expression domain. Specifically, we know state: "Following inhibition of any of the components of these FGFRL-Wnt circuits, the axial expression map shows local shifts in expression domains coincident with the expansion of specific regionally restricted tissues such as brain and pharynx (Figure 7)." The *ndl-3* expression domain shifted in *wntP-2* RNAi animals that had extra pharynges/mouths, which we now clarify in the main text and in figure legends. We do not want to conclude at this point that *wntP-2* directly regulates the *ndl-3* domain, but that the *wntP-2* phenotype involves *ndl-3* expansion. We also performed a new experiment to examine the role of global canonical Wnt signaling (by β-catenin-1 RNAi) in FGFRL expression domains distributions (Figure 2—figure supplement 3). In the progression of the β-catenin-1 RNAi phenotype, we found mild changes in distributions of FGFRL expression domains followed by large-scale re-patterning of the body (ectopic head formation) with associated large-scale FGFRL domain changes. Because of the transformation of the body plan (ectopic heads), it is difficult to make strong conclusions about the latter changes. Overall, we seek to avoid strong conclusions about direct regulation of ndl domains by Wnts, but state that some ndl expression domains change robustly when body plans change following Wnt perturbation.

*5) Levels of WNTP-2, ndl-3, fz5/8-4, ndk RNAi efficacy should be quantified by qRTPCR or RNAseq, as is standard in the planarian field.*

We now included quantification by qRT-PCR as well as FISH experiments to show the efficacy of all the RNAi conditions (Figure 3—figure supplement 1, Figure 5—figure supplement 1).

*6) Because the phenotype of wntA, ndk, and fz5/8-4 are all in a similar direction (head expansion), there cannot be a negative arrow from wntA to ndk in the model in Figure 4. Please modify accordingly if you propose this is a genetic arrow, or increase resolution if you propose this is a small-scale boundary restriction.*

The arrow in the previous version between the wnts and *ndk* was not meant to be a genetic arrow, but rather meant to reflect the expansion of the expression domains of the FGFRLs in Wnt RNAi animals with phenotypes. We modified the presentation of the model such that arrows are not shown anymore; we now simply depict changes in expression domains in 7B and describe them in the legend.

*7) The authors did not put the ndl's in the context of regional head/tail repression by βcat/APC. In the case of the trunk domain expanding posterior, can this be suppressed by βcat RNAi's anteriorizing effects?*

We have now performed this suggested experiment. We inhibited β-catenin-1 with RNAi in animals previously treated with *ndl-3* and *wntP-2* dsRNA. We observed that if anything, β-catenin-1 RNAi further enhances the phenotype observed in the double *ndl-3; wntP-2* RNAi animals (Figure 3—figure supplement 1).

*8) In the end, this is largely an ndl story as well as trying to get a handle on what the muscle cells are capable of in terms of signaling complexity. The authors state that 4/5 ndl's were differentially detected in muscle cells, yet it is hard to get a feeling for how they are regionally restricted, how they overlap with each other, and what how each of their phenotypes affect patterning as a whole. The authors should show the RNAi phenotypes for each of the ndl's and show the effects on the patterning of the others as well as the WNT's used in this manuscript. Secondly, they should quantify the overlap in each of the genes and support this by co-detection in single muscle cells using the data already in hand. The authors missed an opportunity here to perform some analyses on what mRGs were detected in single muscle cells – they should perform such an analysis (instead of the average expression for a region shown in Figure 1). For example, do all ndk-expressing muscles also express wntA and/or fz5/8-4? Are there transcription factor differences that can explain the expression profile of a given muscle? This would be very informative.*

We show the average expression per region and whole-mount in-situ hybridization pattern for all ndls in Figure 1, Figure 1—figure supplement 2, and Figure 2—figure supplement 2 and clarify that all six ndls were differentially expressed in the SCDE analyses (subsection “Regionally expressed genes in muscle, including FGFRL and Wnt-pathway genes, constitute an axial expression map in adult muscle”). All ndls that worked well by FISH (all except ndl-1) were pair-wise tested for spatial overlap (Figure 2). We now analyzed triple FISH for overlap of ndls within individual cells where their spatial domains overlapped, and show this data in a new Figure 2. We also now present heatmaps in the main figure (Figure 2) from single cell sequencing, where individual cell data is shown, allowing visualization of overlap between all ndls in single cells from the anterior regions. We also have commented in the text on the degree of overlap in the single-cell sequencing data (same section). Finally, we show the single-cell sequencing data for all of the muscle cells in Figure 2—figure supplement 2, which shows overlap between mRGs in all single muscle cells spanning the AP axis. In addition, we now analyzed by FISH and quantified by qRT-PCR the expression of each ndl family member when one or more of the ndl genes was inhibited by RNAi (Figure 2—figure supplement 4). We found that single ndl RNAi (and even combination ndl RNAi) only significantly affected the expression of that ndl and not the others in this experiment. We have not identified transcription factor differences that would explain the expression profiles of a given muscle cell, but there are several transcription factors that are mRGs (*zic-1, zic-2, foxD*, prep, sp5, and the Hox genes) most of which have been functionally investigated in other papers.